# Influences on COVID-19 Vaccine Adherence among Pregnant Women: The Role of Internet Access and Pre-Vaccination Emotions

**DOI:** 10.3390/ijerph21060719

**Published:** 2024-05-31

**Authors:** Rosângela Carvalho de Sousa, Maria Juliene Lima da Silva, Maria Rita Fialho do Nascimento, Mayara da Cruz Silveira, Franciane de Paula Fernandes, Tatiane Costa Quaresma, Simone Aguiar da Silva Figueira, Maria Goreth Silva Ferreira, Adjanny Estela Santos de Souza, Waldiney Pires Moraes, Sheyla Mara Silva de Oliveira, Livia de Aguiar Valentim

**Affiliations:** 1Center for Biological and Health Sciences, State University of Pará-UEPA, Campus XII, Santarém 68040-090, PA, Brazil; rosangela.cd.sousa@aluno.uepa.br (R.C.d.S.); m.jsilva@aluno.uepa.br (M.J.L.d.S.); maria.rnascimento@aluno.uepa.br (M.R.F.d.N.); mayara.silveira@aluno.uepa.br (M.d.C.S.); franciane.fernandes@uepa.br (F.d.P.F.); tatiane.quaresma@uepa.br (T.C.Q.); simoneaguiar@uepa.br (S.A.d.S.F.); mariagoreth.ferreira@uepa.br (M.G.S.F.); adjanny.souza@uepa.br (A.E.S.d.S.); sheyla.oliveira@uepa.br (S.M.S.d.O.); 2Institute of Public Health—(ISCO), Federal University of Western Pará (UFOPA), Santarém 68040-255, PA, Brazil; waldiney.moraes@ufopa.edu.br

**Keywords:** fake news, pregnant women, vaccine adherence

## Abstract

Introduction: The onset of the COVID-19 pandemic brought about global uncertainties and fears, escalating the dissemination of fake news. This study aims to analyze the impact of fake news on COVID-19 vaccine adherence among pregnant women, providing crucial insights for effective communication strategies during the pandemic. Methods: A cross-sectional, exploratory study was conducted with 113 pregnant women under care at a Women’s Health Reference Center. Data analysis included relative frequency and odds ratio to assess the relationship between sociodemographic and behavioral variables regarding vaccination. Results: In the behavioral context of vaccination, internet access shows a significant association with decision-making, influencing vaccine refusal due to online information. Nuances in the odds ratios results highlight the complexity of vaccine hesitancy, emphasizing the importance of information quality. Pre-vaccination sentiments include stress (87.61%), fear (50.44%), and anxiety (40.7%), indicating the need for sensitive communication strategies. Discussion: Results revealed that pregnant women with higher education tend to adhere more to vaccination. Exposure to news about vaccine inefficacy had a subtle association with hesitancy, while finding secure sources was negatively associated with hesitancy. The behavioral complexity in the relationship between online information access and vaccination decision underscores the need for effective communication strategies. Conclusions: In the face of this challenging scenario, proactive strategies, such as developing specific campaigns for pregnant women, are essential. These should provide clear information, debunk myths, and address doubts. A user-centered approach, understanding their needs, is crucial. Furthermore, ensuring information quality and promoting secure sources are fundamental measures to strengthen trust in vaccination and enhance long-term public health.

## 1. Introduction

The COVID-19 pandemic, triggered by the emergence of the novel coronavirus (SARS-CoV-2), inaugurated an unprecedented era of uncertainties, fears, and doubts. Initially reported in Wuhan, China, at the end of 2019 [1], the disease rapidly spread globally, leading the World Health Organization (WHO) to declare a pandemic on 11 March 2020. The rapid spread of the virus, combined with the initial absence of specific therapeutic strategies [2,3], propelled the worldwide adoption of stringent control measures, including social distancing, isolation, and mask usage. Amid these efforts, science turned to the accelerated development of vaccines [2,4], a race against time to halt the infection’s progression.

In Brazil, the increase in maternal deaths due to COVID-19 in 2021 highlighted pregnant women as a high-risk group for severe outcomes [3,5]. The immunological changes and physiological adaptations during pregnancy make pregnant women especially vulnerable to respiratory pathogens and severe pneumonias. This global vulnerability directly reflects the specific situation in Brazil, where the rapid development of vaccines, although a scientific feat, generated insecurity and doubts, exacerbated by an avalanche of information often contradictory or false [5,6], increasing vaccine hesitancy among the population, particularly among pregnant women concerned about the potential risks to themselves and their babies.

Vaccine hesitancy, defined as the delay in acceptance or refusal of vaccination despite the availability of vaccination services, is a complex phenomenon influenced by factors such as complacency, convenience, and confidence [7]. The proliferation of misinformation, especially through fake news, has played a significant role in fueling this hesitancy, creating a challenging environment for public health efforts to promote vaccination.

This study aims to investigate the impact of misinformation, particularly fake news, on COVID-19 vaccine adherence among pregnant women. By analyzing the factors that shape vaccination decisions in this vulnerable group, it seeks to contribute crucial insights for developing more effective communication and information strategies. Vaccine hesitancy, amplified by misinformation, not only compromises maternal and infant health but also threatens global efforts to contain the pandemic [5,6]. Therefore, understanding and mitigating its causes among pregnant women is urgent, aiming to strengthen confidence in vaccines and promote broader adherence, thus protecting the most vulnerable populations and contributing to public health more generally.

To this end, aspects such as the influence of internet access on vaccine hesitancy, the main sources of misinformation, and the impact of social media on the spread of fake news will be investigated. The central hypothesis of the study is that misinformation amplifies vaccine hesitancy among pregnant women, with significant implications for public health and pandemic containment.

## 2. Materials and Methods

This cross-sectional, exploratory, analytical-descriptive study employed a quantitative approach to investigate vaccine hesitancy among 113 pregnant women receiving care at a Women’s Health Reference Center. The primary objective was to identify patterns and significant associations between various sociodemographic and vaccination-related variables to understand how these factors might influence attitudes toward vaccination among pregnant women.

### 2.1. Data Collection

Data were collected using a semi-structured questionnaire designed to gather sociodemographic information, including age range, race/ethnicity, education level, marital status, occupation, per capita monthly family income, housing conditions, number of residents per household, internet access, and religious affiliation. Additionally, the questionnaire explored vaccination-related aspects such as knowledge about vaccines, existing doubts, adverse reactions experienced post-vaccination, and emotional responses to vaccination.

### 2.2. Data Organization and Analysis

The collected data were organized and tabulated using Excel^®^, Microsoft 365 MSO (version 2404 Build 16.0.17531.20152) 64 bits, company Microsoft, located in Redmond, WA, USA, which served as a database for quantitative analysis. Statistical analysis was performed using Statistical Package for Social Science (SPSS) version 20.0, 64 bits, for Windows, developed by International Business Machines Corp (IBM), Armonk, NY, USA. The Chi-Square (χ^2^) test was employed to analyze categorical variables and determine significant associations between the studied factors. A *p*-value of less than 0.05 was considered statistically significant, meaning that there is less than a 5% probability that the observed associations occurred by chance. This analysis is shown in Table 1, Table 2 and Table 3.

### 2.3. Inferential Evaluation

The odds ratios (OR) were calculated to provide a nuanced understanding of the relationship between internet access, exposure to online news, difficulty in finding secure sources, and their collective impact on vaccine hesitancy. This analysis was crucial for understanding the complex dynamics influencing pregnant women’s decisions regarding COVID-19 vaccination.

### 2.4. Advanced Analytical Techniques

For the analysis detailed in Table 4, the data were cleaned and transformed, converting categorical variables into dummy variables to facilitate the analysis. Machine learning models, specifically decision trees and random forests, were applied to assess the importance of each variable in predicting trust in misleading information. Hyperparameter tuning was conducted using a simplified approach to optimize model performance, considering technical limitations during the grid search process. The importance of the variables was then calculated and normalized, highlighting the most influential factors. Key variables included the perception that the vaccine can attack the placenta, the contraindication of Coronavac in adolescents, and the belief that the vaccine can cause thrombosis.

### 2.5. Ethical Considerations

Ethical approval for this research was granted by the Research Ethics Committee (CEP) of the State University of Pará, under opinion number 5,727,772. All participants were informed about the study’s aims, and their informed consent was obtained, ensuring adherence to ethical guidelines and the protection of participants’ confidentiality throughout the research process.

### 2.6. Limitations

This study acknowledges potential limitations, such as biases related to self-reported data and the sampling method, which may not represent the broader population. Additionally, technical limitations during hyperparameter tuning may have affected the optimization of machine learning models.

### 2.7. Broader Project Context

This study is part of the broader project titled “Knowledge of Pregnant and Postpartum Women about COVID-19 Vaccination: The Needs of the New Time”, aiming to illuminate critical factors affecting vaccine acceptance among this key demographic. The insights gained from this study are intended to guide the development of targeted interventions to enhance vaccine uptake.

## 3. Results

Table 1 illustrates the distribution of participants by age group, revealing significant concentrations in the 18 to 24 years and 25 to 29 years brackets. This suggests potential distinct characteristics within these segments. The race/ethnicity category shows statistically significant differences, with a notable prevalence of participants identifying as brown. In terms of education, a significant association is observed, with most participants having completed high school, followed by those with higher education. Additionally, marital status reveals significant differences, with a substantial proportion of participants in a stable union. These findings highlight the importance of demographic variables in interpreting responses related to vaccination behaviors and perceptions, contributing to a deeper understanding of the social complexities associated with immunization.

**Table 1 ijerph-21-00719-t001:** Demographic profile of participants: analysis by age group, race/ethnicity, education, and marital Status.

Question	N	%	*p*-Value
**Age Group**			<0.001
15 to 17	6	5.3%	
18 to 24	38	33.61%	
25 to 29	34	30%	
30 to 34	19	16.8%	
35 to 39	12	10.6%	
40 to 44	4	3.53%	
**Ethnicity/Race**			<0.001
White	6	5.3%	
Brown/Mixed Race	91	80.53%	
Black	10	8.84%	
Yellow	4	3.53%	
Not Informed	2	1.76%	
**Education**			<0.001
Elementary School Completed	9	7.96%	
Elementary School Incomplete	10	8.84%	
High School Completed	60	53%	
High School Incomplete	11	9.73%	
Higher Education Completed	16	14.15%	
Higher Education Incomplete	7	6.19%	
**Marital Status**			<0.001
Single	30	26.54%	
Married	26	23%	
Divorced	1	0.88%	
Common-Law Union	56	49.55%	

Source: Authors, 2023.

Table 2 provides an analysis of the behavioral profile concerning vaccination, focusing on education, internet access, and experiences with vaccine doses. The relationship between internet access and decision-making about vaccination is highly significant (*p*-value < 0.001), indicating distinct behaviors among those with internet access compared to those without. The analysis also reveals a significant association between vaccine refusal due to online information (*p*-value < 0.001), underscoring the considerable influence of online sources on vaccination decisions.

The difficulty in finding reliable sources to clarify doubts also shows a significant relationship (*p*-value < 0.001), emphasizing the need for informative and accessible approaches. The analysis of vaccine doses, particularly the first and second doses, shows significant associations, suggesting different factors influencing adherence at these stages. Although the booster dose does not reach statistical significance (*p*-value = 0.059), it still provides insights into behavioral trends related to this vaccination phase. These results underscore the importance of considering the digital context, information sources, and specifics of the vaccination process when developing strategies for awareness and immunization promotion.

**Table 2 ijerph-21-00719-t002:** Behavioral profile regarding vaccination: analysis by internet access and vaccine dose experiences.

Variables	N	%	*p*-Value
**Internet Access**			<0.001
Yes	107	94.69%	
No	6	5.3%	
**Considered not getting vaccinated due to news read on the internet**			<0.001
Yes	78	69.05%	
No	35	30.97%	
**Had difficulty finding reliable sources to clarify doubts**			<0.001
Yes	15	13.27%	
No	95	84.07%	
**Vaccine doses**			
**1st dose**			<0.001
Yes	102	92.03%	
No	11	7.96%	
**2nd dose**			<0.001
Yes	89	78.76%	
No	24	21.23%	
**Booster dose**			0.059
Yes	46	40.7%	
No	67	59.29%	

Source: Authors, 2023.

The interpretation of OR results reveals the relationship between the studied variables and vaccine hesitancy among pregnant women. An OR close to 1.0014 for internet access suggests that having internet access alone is not strongly associated with vaccine hesitancy, indicating that online information access is not a decisive factor in vaccination decisions among the study’s pregnant women.

However, the variables addressing exposure to news about vaccine inefficacy on the internet and difficulty finding secure sources show distinct nuances. The subtle association, with an OR of 1.0013, between exposure to news about vaccine inefficacy and vaccine hesitancy suggests limited influence on vaccination decisions. Conversely, an OR of 0.8087 for difficulty finding secure sources indicates a moderate and negative association, suggesting that pregnant women who find secure sources have a slightly lower probability of hesitating about vaccination.

These results emphasize the complexity of factors influencing vaccine hesitancy, indicating that the relationship between online information access and vaccination decisions is multifaceted. They also highlight the importance of information quality and reliability in influencing decisions, crucial for developing targeted communication strategies and educational interventions to address specific concerns among pregnant women and promote informed vaccination decisions.

Table 3 shows participants’ emotional responses before vaccination, including feelings of happiness, fear, distress, sadness, stress, anxiety, and concern. Notably, 87.61% expressed uncertainty about vaccine efficacy, with immunization causing stress, and 50.44% reported fear, particularly about potential harm to the fetus or complications. Anxiety was mentioned by 40.7% of participants, reflecting significant emotional impacts associated with vaccination.

**Table 3 ijerph-21-00719-t003:** Feelings and reactions towards immunization.

Question	n	%	*p*-Value
**What feelings did you experience before getting vaccinated?**			<0.001
Happiness	8	7.07%	
Fear	57	50.44%	
Anguish	11	9.73%	
Sadness	11	9.73%	
Stress	99	87.61%	
Anxiety	40	40.7%	
Concern	22	23.00%	

Source: Authors, 2023.

Table 4 analyzes factors influencing trust in false news statements related to the pandemic. The most influential variable is the perception that the vaccine can attack the placenta (P11), with a normalized importance of 25.97%, highlighting concerns about vaccines’ potential impacts on reproductive health. Another significant variable is the contraindication of Coronavac in adolescents (P7), with a normalized importance of 9.93%, reflecting concerns about vaccine safety for younger age groups. The belief that the vaccine can cause thrombosis (P3) is also relevant, with an importance of 9.78%, reflecting widespread fears.

**Table 4 ijerph-21-00719-t004:** Factors influencing trust in false news statements.

Feature	Importance	Variable Type	Normalized Importance
Perception that the vaccine can attack the placenta	0.085548	Categorical	0.25967
Contraindication of Coronavac in adolescents	0.032730	Categorical	0.09930
Belief that the vaccine can cause thrombosis	0.032226	Categorical	0.09781
Adverse reactions to the Coronavac booster dose	0.028399	Categorical	0.08619
Marital status (category 2)	0.027824	Categorical	0.08445
Education level (category 3)	0.025022	Categorical	0.07597
Religion (category 3)	0.024815	Categorical	0.07534
Belief that those who take the vaccine may show COVID-19 symptoms	0.024411	Categorical	0.07408
Vaccine used (category 2)	0.024333	Categorical	0.07383
Belief that pregnant women may suffer miscarriage after vaccination	0.024213	Categorical	0.07347

Source: Authors, 2023.

The adverse reactions to the Coronavac booster dose (P8) are another notable factor, with an importance of 8.62%. This demonstrates that reports of side effects can significantly influence public perception. The marital status of respondents also plays a role, with an importance of 8.45%, suggesting that marital status may be correlated with different levels of trust in false news, possibly due to underlying social or economic variables.

Education level appears with an importance of 7.60%. This finding underscores that educational level can affect the ability to discern between true and false information. Similarly, religiosity has an importance of 7.53%, indicating that religious beliefs can influence receptiveness to erroneous information.

The belief that those who receive the vaccine may exhibit COVID-19 symptoms (P5) is also significant, with an importance of 7.41%. This type of misinformation can cause unfounded concerns about the efficacy and safety of vaccines. Finally, the variable related to the vaccine used and the belief that pregnant women may suffer miscarriages after vaccination (P10) complete the list of the most influential variables, both with importance around 7.38% and 7.35%, respectively.

In summary, perceptions of adverse vaccine effects play a crucial role in shaping trust in false news. Factors such as marital status, educational level, and religious beliefs also show significant influence. These findings highlight the complexity of the influences on trust in health-related information and underscore key areas where education and effective communication can help combat misinformation.

## 4. Discussion

The results of our study demonstrate a significant impact of fake news on vaccine hesitancy among pregnant women, highlighting the pivotal role of internet access in influencing vaccination decisions. The findings reveal that 94.69% of the participants had internet access, which, while facilitating access to valuable vaccine information, also exposed them to fake news, leading to vaccine hesitancy in 40.7% of respondents. This underscores the importance of digital literacy and the development of critical thinking skills as crucial interventions. Similar to the conclusions drawn by Passos and Filho [7], our study emphasizes that education significantly affects vaccine acceptance. Our findings align with those of Silva et al. [2], suggesting that higher education levels correlate with increased knowledge about diseases and preventive strategies, thus reducing vaccine hesitancy.

The observed regression in vaccine uptake from the first dose to subsequent doses suggests a complex interaction of factors including misinformation, logistical challenges, and a potential lack of targeted communication efforts. This complexity mirrors the broader literature that identifies misinformation as a significant driver of vaccine hesitancy, necessitating multifaceted strategies to counteract this trend [8,9,10]. Proposed strategies to address these challenges include enhancing digital literacy among pregnant women to empower them to discern reliable information sources and developing targeted communication campaigns that specifically address their concerns, debunk myths, and clarify doubts about vaccine safety and efficacy. These strategies underscore the need for leveraging multiple platforms, including social media, to effectively reach this audience [11].

Furthermore, strengthening communication skills among healthcare providers to address pregnant women’s fears and concerns can provide the reassurance needed to make informed vaccination decisions. This approach is critical in light of the nuanced impact of internet access and exposure to misinformation on vaccine hesitancy observed in our study. While internet access alone was not a strong predictor of hesitancy, the exposure to misinformation significantly influenced vaccination decisions. This finding underscores the urgent need for ensuring the reliability and quality of information accessible to pregnant women, aligning with broader discussions on the impact of fake news on public health [6,10].

The high levels of stress, fear, and anxiety reported by participants underscore the importance of addressing the informational and emotional dimensions of vaccine decision-making. Interventions that create safe spaces for discussions, offer counseling, and develop supportive communities can be crucial in mitigating vaccine hesitancy. These measures can help in navigating the emotional landscape that accompanies vaccination decisions, thereby supporting pregnant women in making informed choices.

The analysis presented is reinforced by the understanding of the cognitive determinants of vaccine hesitancy among pregnant women, further emphasizing the critical role of education and digital literacy [12]. In light of these findings, it becomes evident that targeted strategies to increase awareness and discernment regarding online vaccine information are indispensable. The study [12] corroborates our observation on the significant influence of exposure to fake news on vaccine hesitancy and the importance of multifaceted strategies to mitigate this effect. This includes not only improving digital literacy among the targeted audience but also implementing communication campaigns that specifically address their concerns, clarify doubts, and proactively debunk myths. Furthermore, the commitment to strengthening the communication skills of healthcare providers, as suggested by our study, is essential to ensure that pregnant women receive the necessary support to make informed decisions.

The analysis of trust in false news related to the pandemic underscores the multifaceted nature of public perception and the significant role of misinformation. A key factor influencing trust is the concern about vaccine safety, particularly in younger populations. The contraindication of Coronavac in adolescents reflects a broader anxiety about administering vaccines to this age group. This concern is likely exacerbated by the rapid spread of misinformation, which can amplify fears and lead to greater distrust in health recommendations [13,14].

Another prominent issue is the belief that vaccines can cause severe side effects, such as thrombosis [15]. This concern has been widely discussed and has permeated public discourse, illustrating how even well-publicized but rare adverse events can shape public opinion. The impact of reported side effects, particularly from booster doses, further demonstrates how isolated incidents can be magnified, affecting overall trust in vaccination programs. These perceptions highlight the need for transparent and context-rich communication from health authorities to mitigate unwarranted fears.

Marital status emerges as a variable that influences trust in false news, suggesting that personal and social circumstances may affect susceptibility to misinformation. Married individuals or those in different marital arrangements might experience varying levels of exposure to reliable information or may be differently influenced by their social networks. This points to the broader social dynamics at play in the dissemination and acceptance of health information.

Education level is another critical factor. Individuals with higher educational attainment are generally better equipped to evaluate the credibility of information sources and to distinguish between accurate and misleading information [16]. This reinforces the importance of educational initiatives aimed at improving media literacy and critical thinking skills as tools to combat misinformation.

Religiosity also plays a role in shaping receptiveness to false information. Trust in religious leaders and communities can sometimes outweigh trust in scientific sources, especially when there is conflicting information [17]. This underscores the need for engaging with religious communities and leaders in public health communication strategies to ensure that accurate information reaches diverse audiences.

The belief that vaccinated individuals may exhibit COVID-19 symptoms is indicative of the kind of misinformation that can undermine public confidence in vaccines. Such beliefs can lead to vaccine hesitancy, fueled by unfounded concerns about vaccine efficacy and safety. Additionally, ongoing fears about the effects of vaccines on pregnancy highlight persistent misinformation that needs to be addressed through targeted, empathetic, and evidence-based communication.

This study acknowledges several limitations that could influence the results and their interpretation. One key limitation is the reliance on self-reported data, which may introduce biases such as social desirability bias, where participants might provide responses they perceive as more acceptable. Additionally, the sample size of 113 pregnant women, while providing valuable insights, may not be fully representative of the broader population of pregnant women in Brazil. This limits the generalizability of the findings. The use of a semi-structured questionnaire, although comprehensive, may have constrained participants’ responses, potentially omitting other relevant factors influencing vaccine hesitancy. Moreover, the cross-sectional design of the study captures data at a single point in time, which limits the ability to establish causality between the identified factors and vaccine hesitancy. Future research should consider longitudinal designs to better understand the temporal dynamics of vaccine hesitancy and the evolving impact of misinformation.

The strong influence of misinformation, particularly regarding the potential adverse effects of vaccines, underscores the need for targeted communication strategies that address specific concerns and misinformation prevalent among pregnant women. Public health campaigns should leverage trusted sources and community leaders to disseminate accurate and reliable information about vaccine safety and efficacy. Additionally, improving internet access and digital literacy can empower pregnant women to navigate online information more effectively, reducing the impact of misleading content. Educational interventions should be designed to enhance critical thinking skills and provide clear, evidence-based information to counteract the pervasive influence of fake news. By addressing these issues, public health initiatives can foster a more informed and confident approach to vaccination, ultimately improving health outcomes for pregnant women and their infants.

Overall, the analysis reveals the complex interplay of factors influencing trust in health-related information. Effective public health strategies must consider these diverse influences, leveraging education, transparent communication, and community engagement to build and maintain public trust. Addressing misinformation requires a multifaceted approach that considers social, educational, and cultural dimensions to foster a well-informed and resilient public. By implementing targeted educational interventions, enhancing digital literacy, and fostering supportive environments for informed decision-making, we can aim to improve vaccine uptake among this vulnerable population. This concerted effort will not only protect maternal and fetal health in the context of the COVID-19 pandemic but also contribute to the broader goal of enhancing public health resilience against misinformation and its impacts.

## 5. Conclusions

The results highlight the complexity of vaccine hesitancy among pregnant women, revealing that online information access, although widespread, is not a decisive determinant. The association between exposure to news about vaccine inefficacy and hesitation is subtle, while the difficulty in finding reliable sources shows a more moderate and negative association. To address the challenges of vaccine hesitancy among pregnant women, a comprehensive and proactive approach is necessary. The development of specific campaigns targeting this audience emerges as a crucial strategy. Such campaigns should be meticulously designed to address the specific concerns identified in the study, providing clear and accurate information about vaccines, debunking prevalent myths, and clarifying doubts that may influence the decision to adhere to vaccination.

Moreover, close collaboration with online platforms becomes imperative, requiring the implementation of rigorous policies to monitor, control, and reduce the spread of fake news, thereby mitigating the dissemination of incorrect information contributing to vaccine hesitancy. By focusing not only on increasing information access but also ensuring its quality and reliability, promoting secure and verified sources, it is possible to build a solid foundation of trust.

In addition to the proposed strategies, it is essential to investigate how specific cultural, social, and emotional dynamics among pregnant women influence their perceptions and vaccination decisions. Future research should focus on the effectiveness of customized communication strategies, assessing their impact across different contexts and utilizing a variety of message formats. Analyzing the role of healthcare professionals as trusted intermediaries of vaccine information and their influence on pregnant women’s vaccination decisions also deserves attention. Furthermore, developing interventions to enhance digital literacy among expectant mothers could be key to minimizing the effects of misinformation. Studies that involve pregnant women in the design of educational materials and communication strategies ensure that interventions are not only relevant but also deeply resonate with their experiences and concerns. Such research efforts are crucial for developing more effective approaches to mitigate vaccine hesitancy and promote public health more broadly, thereby ensuring that future generations benefit from comprehensive and reliable vaccination coverage.

## Data Availability

The data are not available due to ethical and privacy restrictions; researchers may request through the Research Ethics Committee of the State University of Pará via email: cepuepa@outlook.com.

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
