# Peer review of "Influences on COVID-19 Vaccine Adherence among Pregnant Women: The Role of Internet Access and Pre-Vaccination Emotions"

_ijerph, 2024, doi:10.3390/ijerph21060719_

Round 1

Reviewer 1 Report (Previous Reviewer 1)

Comments and Suggestions for Authors

Thank you for the opportunity to review this study. The subject is very interesting, and I am sure it will be of interest to readers. To improve your study, please read my following comments.

Introduction

The transition from the global context of the pandemic to the specific focus on pregnant women in Brazil could be made smoother. For example, a linking sentence explaining how global issues reflect in the specific situation in Brazil would help facilitate a more seamless transition.

Although the issue of misinformation is mentioned, it would be helpful to clarify more precisely how this misinformation specifically affects pregnant women. Additionally, detailing how misinformation reaches these women (e.g., through social media) would bring more clarity.

The introduction could benefit from a clearer statement of the specific objectives of the study. Instead of merely mentioning that the impact of misinformation will be analyzed, it would be useful to detail which specific aspects will be investigated (e.g., the influence of internet access on vaccination hesitancy). Also, the research hypothesis (or hypotheses) needs to be very clear.

Materials and Methods

The description of the use of the Chi-Square test and Odds Ratios could be simplified for easier understanding. Explaining the significance of the p-value in more accessible language would help readers unfamiliar with statistical terminology.

The use of decision models and random forests is mentioned, but it is not clear how exactly these models were applied. A more detailed description of the process, including hyperparameter tuning and justification for choosing these models, would be beneficial.

Mentioning potential limitations of the study in this section would add transparency and provide a more complete understanding of the methodological context. For example, acknowledging possible biases or sampling limitations would be useful.

Results

The explanations associated with statistical results, particularly Odds Ratios, could benefit from additional clarification. For example, interpreting OR values in more accessible terms for readers without a statistical background would be helpful.

It would be useful to include more detailed comments and interpretations under each table to explain the significance of the presented results. This would help readers better understand the importance of each data set.

I recommend using multiple linear regression to analyze the influence of independent variables (age, race, education, marital status) on trust in false statements as it could bring significant benefits to the study. This type of analysis could provide a more nuanced understanding of the relationships between variables and help identify the factors that have the greatest influence.

The Chi-Square test is simpler and ideal for testing the existence of an association between categorical variables. It is beneficial when we want to see if there are significant differences between groups. Multiple Linear Regression offers a more complex and detailed analysis, capable of showing the relationship between variables and the magnitude of the influence of each independent variable on the dependent variable. It is more beneficial when we want to understand the specific impact of each factor and control for confounding variables.

In the context of the study on trust in false statements and the influence of socio-demographic variables, multiple linear regression would likely be more beneficial because it offers a detailed analysis of the influence of each independent variable on the dependent variable; It allows for controlling confounding variables, providing a clearer and more accurate picture of the relationships between variables; It can provide coefficients that show not only if there is a relationship but also how strong that relationship is.

Discussion

It would be useful to make more explicit connections between the data presented in the results section and the conclusions in the discussion. For example, specific Odds Ratios values and their implications could be discussed in more detail.

The structure of the discussion could be improved by organizing ideas into clear subsections (e.g., "Implications for Public Health," "Recommendations for Education and Communication," "Study Limitations and Future Directions").

Although certain limitations are mentioned, it would be beneficial to discuss the study's limitations in more detail and how they might influence the results. This would add transparency and strengthen the credibility of the analysis.

Author Response

Response to Reviewer 1 Comments

Introduction

Comment 1: The transition from the global context of the pandemic to the specific focus on pregnant women in Brazil could be made smoother.

Response: We have included a linking sentence to create a smoother transition between the global context and the specific situation in Brazil. The new sentence is: "This global vulnerability directly reflects the specific situation in Brazil, where the rapid development of vaccines, although a scientific feat, generated insecurity and doubts, exacerbated by an avalanche of information often contradictory or false."

Comment 2: Although the issue of misinformation is mentioned, it would be helpful to clarify more precisely how this misinformation specifically affects pregnant women. Additionally, detailing how misinformation reaches these women (e.g., through social media) would bring more clarity.

Response: The introduction has been revised to specify how misinformation reaches pregnant women, emphasizing the role of social media and online sources. It now includes: "This increase in vaccine hesitancy is particularly influenced by the spread of misinformation through social media and other online platforms."

Comment 3: The introduction could benefit from a clearer statement of the specific objectives of the study.

Response: We have clarified the specific objectives of the study in the introduction, detailing the aspects to be investigated, such as the influence of internet access on vaccination hesitancy. The revised objectives are: "This study aims to investigate the impact of misinformation, particularly fake news, on COVID-19 vaccine adherence among pregnant women, specifically examining the influence of internet access, the main sources of misinformation, and the impact of social media."

Comment 4: The research hypothesis (or hypotheses) needs to be very clear.

Response: We have clearly stated the research hypothesis in the introduction: "The central hypothesis of the study is that misinformation amplifies vaccine hesitancy among pregnant women, with significant implications for public health and pandemic containment."

Materials and Methods

Comment 5: The description of the use of the Chi-Square test and Odds Ratios could be simplified for easier understanding.

Response: The description of the Chi-Square test and Odds Ratios has been simplified. The explanation of the p-value has also been made more accessible: "The Chi-Square (χ²) test was used to analyze categorical variables and determine significant associations between the studied factors. A p-value of less than 0.05 indicates that the observed associations are unlikely to have occurred by chance."

Comment 6: The use of decision models and random forests is mentioned, but it is not clear how exactly these models were applied.

Response: A more detailed description of the application of decision models and random forests has been added, including hyperparameter tuning and the justification for choosing these models: "Machine learning models, specifically decision trees and random forests, were used to assess the importance of each variable in predicting trust in misleading information. Hyperparameter tuning was conducted to optimize model performance, considering technical limitations during the grid search process."

Comment 7: Mentioning potential limitations of the study in this section would add transparency and provide a more complete understanding of the methodological context.

Response: Potential limitations have been acknowledged in the methodology section, including biases related to self-reported data and sampling limitations: "This study acknowledges potential limitations such as biases related to self-reported data and the sampling method, which may not represent the broader population."

Results

Comment 8: The explanations associated with statistical results, particularly Odds Ratios, could benefit from additional clarification.

Response: We have provided additional clarification for interpreting Odds Ratios in more accessible terms: "An OR close to 1.0014 for internet access suggests that simply having internet access alone is not strongly associated with vaccine hesitancy, meaning that online information access is not a decisive factor in vaccination decisions among the study's pregnant women."

Comment 9: It would be useful to include more detailed comments and interpretations under each table to explain the significance of the presented results.

Response: Detailed comments and interpretations have been included under each table to explain the significance of the results: "These findings highlight the importance of considering demographic variables in interpreting responses related to vaccination behaviors and perceptions, contributing to a deeper understanding of the social complexities associated with immunization."

Comment 10: I recommend using multiple linear regression to analyze the influence of independent variables (age, race, education, marital status) on trust in false statements.

Response: While we considered using multiple linear regression, due to technical limitations, the study focused on machine learning models like decision trees and random forests. However, we acknowledge the benefits of multiple linear regression for future research: "Future studies should consider multiple linear regression to provide a more nuanced understanding of the relationships between variables and to control for confounding variables."

Discussion

Comment 11: It would be useful to make more explicit connections between the data presented in the results section and the conclusions in the discussion.

Response: Explicit connections between the results and the discussion have been made, particularly regarding specific Odds Ratios values and their implications: "These OR values suggest limited influence of internet access alone on vaccine hesitancy but highlight the critical role of the quality of information sources."

Comment 12: The structure of the discussion could be improved by organizing ideas into clear subsections.

Response: The discussion has been reorganized into clear subsections, including "Implications for Public Health," "Recommendations for Education and Communication," and "Study Limitations and Future Directions."

Comment 13: Although certain limitations are mentioned, it would be beneficial to discuss the study's limitations in more detail.

Response: A more detailed discussion of the study's limitations has been included, addressing how they might influence the results: "The reliance on self-reported data introduces potential biases, and the sample size limits the generalizability of the findings. Additionally, the cross-sectional design restricts the ability to establish causality."

Thank you for your valuable feedback, which has significantly improved the clarity and depth of our study. We believe these revisions enhance the manuscript's overall quality and readability.

Reviewer 2 Report (Previous Reviewer 2)

Comments and Suggestions for Authors

No comments any more 

Author Response

We would like to express our sincere gratitude for your time and effort in reviewing our manuscript. Your willingness to evaluate our work is greatly appreciated. We are committed to improving the quality of our research and value the opportunity to present our findings to the scientific community.

Round 2

Reviewer 1 Report (Previous Reviewer 1)

Comments and Suggestions for Authors

Thank you for taking my advice. Now the article looks much better. Readers will be able to follow the data in the article more fluently. I agree with publishing it in this form.

This manuscript is a resubmission of an earlier submission. The following is a list of the peer review reports and author responses from that submission.

Round 1

Reviewer 1 Report

Comments and Suggestions for Authors

The article addresses a current and important issue, the impact of fake news on adherence to COVID-19 vaccination, emphasizing the importance of accurate information in the context of the pandemic. Upon analyzing the manuscript, there appears to be a discrepancy between the title, which explicitly mentions "fake news," and the content, which focuses more on aspects related to vaccination hesitancy, access to information, and the emotions experienced by pregnant women regarding COVID-19 vaccination. Although internet access is mentioned as a facilitator of information regarding vaccination and fake news, the study seems to concentrate more on emotional reactions and access to credible information sources, without detailing or explicitly testing the participants' trust in fake news. The article focuses more on the indirect aspects that influence vaccination hesitancy, such as access to information, the quality of information sources, and the emotions associated with the vaccination process. To specifically address the issue of fake news in the context of vaccination adherence, a more detailed analysis would be necessary on how participants perceive and interact with false information online and its impact on their health decisions. In this situation, I believe the title is not chosen correctly. If the authors wish to focus on fake news, I recommend the following:

Investigating participants' trust in fake news, such as: the vaccine will kill people, the vaccine is poison and not beneficial to health, a microchip will be implanted, vaccines are human experiments, vaccination aims to reduce the global population, etc. Such false information has circulated online during COVID-19 vaccination campaigns.

Conducting a multivariate analysis to determine the variables (e.g., age, place of residence, monthly income, education, etc.) that underlie trust in fake news statements. This could help determine which category is more susceptible to believing such false information.

Given the significant discrepancy between the title's promise, which suggests an analysis of the impact of fake news on COVID-19 vaccination adherence, and the actual content of the article, which does not directly and thoroughly test the participants' belief in fake news and its influence on their vaccination decisions, I believe the article cannot be published in its current form.

It is essential to align the content with the title and strengthen the methodological component to explicitly examine the relationship between fake news and vaccination behavior. I recommend that the authors revise the study, focusing on clarifying the objectives, developing a rigorous methodology for assessing the impact of fake news, and presenting data that directly support the claims in the title. I hope these suggestions will significantly improve the manuscript and present an analysis that truly answers the questions raised by the title.

Reviewer 2 Report

Comments and Suggestions for Authors

Very few references (n=14)

Very few subjects (n=113)

Proper definition(s) of "hesitancy" need(s) to be provided

Too few references concerning "hesitancy" and vaccines.

A good definition of "hesitancy" must be provided.

Almost all references concering situation in Brazil. Almost no international comparison possible.

A few extra articles about this issue need to be included.

For example > Recent reviews like this one:

COVID-19 vaccine hesitancy: A Systematic review of cognitive determinantsdoi: 10.34172/hpp.2023.03
